


# Incorporation of inline warm-rain diagnostics into the COSP2 satellite simulator for process-oriented model evaluation

Takuro Michibata[1], Kentaroh Suzuki[2], Tomoo Ogura[3], and Xianwen Jing[2]

[1]Research Institute for Applied Mechanics, Kyushu University, Fukuoka, Japan
[2]Atmosphere and Ocean Research Institute, University of Tokyo, Chiba, Japan
[3]National Institute for Environmental Studies, Ibaraki, Japan

**Correspondence:** Takuro Michibata (michibata@riam.kyushu-u.ac.jp)

**Abstract.** Cloud Feedback Model Intercomparison Project Observational Simulator Package (COSP) has been widely used to diagnose model performance and physical processes via an apple-to-apple comparison to satellite measurements. Although the COSP provides useful information about clouds and their climatic impact, outputs that have a subcolumn dimension require large amounts of data. This can cause a bottleneck when conducting sets of sensitivity experiments or multiple model intercom-
5 parisons. Here, we incorporate two diagnostics for warm-rain microphysical processes into COSP2, the latest version of the simulator. The approach used here employs existing diagnostic methodologies that probe how the warm-rain processes occur using statistics constructed from simulators of multiple satellite instruments along with their subcolumn information. The new diagnostics are designed to produce statistics online during the COSP execution, eliminating the need to output subcolumn variables. Users can also readily conduct regional analysis tailored to their particular research interest (e.g., land–ocean differ-
10 ences), using an auxiliary post-process package after the COSP calculation. This inline tool also generates global maps of the occurrence frequency of warm-rain regimes (i.e., non-precipitating, drizzling, and precipitating) classified according to Cloud-Sat radar reflectivity, putting the warm-rain process diagnostics into the context of geographical distributions of precipitation. The inline diagnostics are applied to the MIROC6 GCM to demonstrate how known biases common among multiple GCMs relative to satellite observations are revealed. The inline multisensor diagnostics are intended to serve as a tool that facilitates
process-oriented model evaluations in a manner that reduces the burden on modelers for their diagnostics effort.

## 1 Motivation

Clouds play a critical role in the global climate system by controlling the hydrological cycle and radiation budget (Wood, 2012; L'Ecuyer et al., 2015; Matus and L'Ecuyer, 2017). However, general circulation models (GCMs) still contain large
uncertainties related to cloud processes associated with subgrid-scale parameterizations, cloud feedbacks, and microphysics (Bretherton, 2015; Gettelman and Sherwood, 2016; Mülmenstädt and Feingold, 2018). In particular, modeling aerosol–cloud interactions remains challenging (Boucher et al., 2013; Myhre et al., 2013) because warm-rain processes, which are central



to the the aerosol–cloud interactions of low clouds, are highly sensitive to aerosols (e.g., Quaas, 2015) and are also regime dependent (Medeiros and Stevens, 2011; Gryspeerdt and Stier, 2012; Michibata et al., 2016; Bai et al., 2018).

Global satellite observations, particularly those of satellite constellations, are a powerful tool (e.g., Stephens et al., 2018) that can be used to improve GCM parameterizations by constraining aerosol–cloud relationships (Wang et al., 2012; Suzuki
et al., 2013). However, direct comparisons between native model output and satellite-retrieved data are not always straightforward ("apple-to-orange" comparisons), because satellite retrievals are inverse estimates from observed radiance or radar reflectivity factor (e.g., Masunaga et al., 2010). Therefore, native model values must be converted by solving the "forward problem" using the same algorithms applied to each satellite sensor for consistent ("apple-to-apple") comparisons. To this end, the Cloud Feedback Model Intercomparison Project (CFMIP) community has developed the CFMIP Observation Simulator
Package (COSP; Bodas-Salcedo et al., 2011), which provides "a common language for clouds" (Swales et al., 2018). With this capability, COSP has been used widely, not only in the CFMIP community, but by many climate modelers, to evaluate model uncertainties through model intercomparisons, such as CMIP6 (Eyring et al., 2016; Webb et al., 2017).

The current version of the simulator package comprises the ISCCP (Klein and Jakob, 1999; Webb et al., 2001), MODIS (Pincus et al., 2012), MISR (Marchand and Ackerman, 2010), PARASOL (Konsta et al., 2016), CloudSat (Haynes et al.,
2007), and CALIPSO (Chepfer et al., 2008; Cesana and Chepfer, 2012) simulators. To effectively utilize these capabilities, there is a growing need for "process-oriented" model diagnostics, which have been recognized as essential to the community effort to advance climate modeling (Tsushima et al., 2017; Webb et al., 2017). To fulfill this need, the COSP package must be continually optimized for efficiently production of process diagnostics.

The recent and significant redesign of COSP aimed to provide more robust and efficient code (Swales et al., 2018). The
20 updated package (COSP2) enhances the flexibility by allowing for native model subgrid cloud representations to be used as input for the COSP2 interface. Using inputs from a host model, simulators in COSP2 perform two main tasks (Fig. 1): 1) translating the native model variables to subcolumn (pixel) scale synthetic retrievals, and 2) aggregating the subcolumn retrievals to column (grid) scale statistics (see Fig. 1 of Swales et al. (2018) for details). This substantial revision of COSP has extended its functionality, enabling the introduction of diagnostics constructed from multiple instrument simulators.

To investigate microphysics at a fundamental process-level, it is best to analyze the instantaneous output for the variables of interest rather than their monthly means (e.g., Konsta et al., 2016). This is because these processes typically occur over short timescales ("fast processes") and contribute to the regime dependency of important phenomena including aerosol–cloud–precipitation interactions (Michibata et al., 2016). This requires high-frequency data output (∼6 hourly) from COSP (see also Table 1 of Tsushima et al. (2017)), which results in large amounts of data, particularly when subcolumn (pixel scale)
variables, such as the radar or lidar simulators are involved. The recommendation to COSP users is to assume approximately 100 subcolumns per 1° of model grid spacing. This leads to bottlenecks in fast process diagnostics that analyze instantaneous output in terms of both data transfer and analysis.

To address this challenge in COSP, this work incorporates an inline diagnostic tool into COSP2 to facilitate process-oriented model evaluations targeted at warm-rain. By introducing joint statistics from multiple satellite simulators, detailed information
related to cloud microphysics is now readily available from model diagnostics without the need to output subcolumn variables.





Although this tool is applied here to warm-rain diagnostics, it can be extended to other microphysical processes to facilitate the efficient evaluation of models with subgrid cloud schemes of various complexity (Turner et al., 2012; Thayer-Calder et al., 2015; Tompkins and Di Giuseppe, 2015; Norris and da Silva, 2016; Griffin and Larson, 2016; Ovchinnikov et al., 2016).

This technical paper is organized as follows: the diagnostic tool that is based on the joint satellite simulators and its application to model evaluations are described in section 2; the scientific perspectives using the warm-rain diagnostic tool and A-Train satellite data are provided in section 3; and a summary and future work are presented in section 4. The source codes and reference satellite data are all available from public repositories (see 'Code and data availability' below).

## 2   Concept and design

### 2.1   Warm-rain diagnostics

A radar–height histogram, the so-called contoured frequency by altitude diagram (CFAD), is a default output from the Cloud-Sat radar simulator in COSP (Bodas-Salcedo et al., 2011) and describes the vertical profile of hydrometeors. Although the CFAD provides complete profile statistics including all types of hydrometeors (i.e., liquid droplets, ice crystals, raindrops, and snowflakes), more specific statistics are often useful when investigating a particular process, including the warm-rain processes that are the focus of this work. One of the transformative advances made possible by combining active and passive satellite measurements is the ability to generate observational diagnostics of variations in the microphysical vertical structure of
clouds caused by the surrounding environment, such as aerosol concentration and dynamical regimes (Marchand et al., 2009; Sorooshian et al., 2013; Nam et al., 2014; Christensen et al., 2016; Ma et al., 2018; Rosenfeld et al., 2019). Such diagnostics are made possible by combining multiple satellite observations to construct joint statistics that "fingerprint" the process of interest. For this study, we incorporated two such diagnostics based on the CloudSat and MODIS satellite simulators into COSP2 to
evaluate cloud-to-rain microphysical transition processes represented in GCMs using satellite observations. Both diagnostics are applied only to single-layer warm clouds (SLWCs) and their results are constructed with the aid of the column simulators, as illustrated in Fig. 1.

The first diagnostic provides the fractional occurrence of warm-rain regimes, which are classified according to the CloudSat column maximum radar reflectivity ($Z_{\max}$) as non-precipitating ($Z_{\max} < -15$), drizzling ($-15 < Z_{\max} < 0$ ), and precipitat-
ing ($0 < Z_{\max}$). The occurrence frequencies of the non-precipitating, drizzling, and precipitating regimes are defined at the pixel-scale as:

$$f_i(\lambda, \phi) = \frac{n_i(\lambda, \phi)}{n_{\mathrm{slwc}}(\lambda, \phi)} \tag{1}$$

where $i \in \{\text{cloud, drizzle, rain}\}$, and $n_{\mathrm{slwc}}$ is the total sample number of the SLWCs detected by CloudSat and MODIS retrievals within the grid box at longitude $\lambda$ and latitude $\phi$. This metric provides information about where and how the warm-
rain occurrence frequency and intensity are biased in the model relative to the satellite observations (Jing et al., 2017; Kay et al., 2018).





The second diagnostic is the probability density function (PDF) of radar reflectivity profiles scaled as a function of the vertically sliced in-cloud optical depth (ICOD), and is commonly referred to as the contoured frequency by optical depth diagram (CFODD), as proposed by Nakajima et al. (2010) and Suzuki et al. (2010). The diagnostic reveals how the vertical microphysical structures of SLWCs tends to transition from non-precipitating to precipitating regimes as a fairly monotonic function of the cloud-top particle size. In this method, the MODIS-retrieved columnar cloud optical depth ($\tau_c$) is redistributed into a layered ICOD at each radar height ($h$) bin, according to the adiabatic-condensation growth model (Brenguier et al., 2000; Szczodrak et al., 2001; Bennartz, 2007) as:

$$\mathrm{ICOD}(h) = \tau_c \left[ 1 - \left( \frac{h}{H} \right)^{5/3} \right] \qquad (2)$$

where $H$ is the cloud geometric thickness. After scaling by ICOD (optical depth from the cloud-top), the CFODD reveals particle coalescence processes (Suzuki et al., 2010) and offers a direct way to evaluate and constrain these processes in global models (Suzuki et al., 2011, 2015).

The A-Train analysis compared with the model statistics is also restricted to SLWCs, which are defined as having cloud-top temperatures ($\mathrm{T_{top}}$) > 273.15 K, extracted using the CloudSat radar reflectivity and a cloud mask described by Michibata et al. (2014, 2016). Convective deep clouds are thus excluded from the analysis. To ensure consistency with A-Train observations, both diagnostics for GCMs/COSP2 use only subcolumn pixels with a scene type of stratiform clouds (fracout = 2), as shown in Fig. 1.

## 2.2 Computational procedure and outputs

The warm-rain diagnostics (occurrence frequency of warm-rain regimes and CFODD) are activated by setting the logical flags "Lwr_occfreq" and "Lcfodd" to *true* in the output namelist (cosp_output_nl_v2.0.txt). Both the CloudSat and MODIS simulators are included automatically in the calculations if either flag is set to *true*, and the specified diagnostics are generated (see Fig. 1) during COSP execution.

The generated outputs are the total number of samples in each grid-box ($\lambda$,$\phi$), which are aggregated from the subcolumn retrievals. These outputs were chosen because the diagnosed PDFs should be created by using total samples during the course of simulation. Because this requires a post-processing of the output to construct the statistics, a post-processing package is also prepared to support this procedure. The post-processing package also facilitates regional analysis tailored to a users' particular research purpose, as discussed later. Users are recommended to output the diagnostics as an accumulated value (e.g., for each month) rather than instantaneous values, to reduce the volume of output data.

## 3 Analysis examples and scientific perspectives

We used the MIROC6-SPRINTARS global aerosol–climate model (e.g., Tatebe et al., 2018) to demonstrate the warm-rain analysis of the diagnostic tool. The host model resolution was $1.4° \times 1.4°$ with 40 vertical levels (T85L40). The numbers of subcolumns (NCOLUMNS) was set to 140. The model time step was 12 min, and COSP was called every 3 hr. The COSP2





simulator was operated for one full year after a one-year spin-up. Simulations were conducted under climatological sea-surface temperature and sea ice, present-day aerosol emissions, and greenhouse gases with monthly mean annual cycles. A benchmark test indicated that the inline warm-rain diagnostic tool increases the computational cost by only about 0.8% when using the SX-ACE supercomputer system of the National Institute for Environmental Studies, Japan.

As a reference, we also calculated the target metrics (i.e., the occurrence frequency of SLWCs and CFODDs) using CloudSat and MODIS satellite data products (e.g., Polonsky, 2008; Mace et al., 2007; Marchand et al., 2008; Partain, 2007; Stephens et al., 2008) for the period June 2006–April 2011. Detailed descriptions of the MIROC-SPRINTARS GCM and A-Train products are provided elsewhere (Watanabe et al., 2010; Michibata and Takemura, 2015; Tatebe et al., 2018; Michibata et al., 2019; Stephens et al., 2008, 2018).

**3.1   Occurrence frequency of warm clouds**

Figure 2 shows geographical distributions of SLWCs and their fractional occurrences for non-precipitating, drizzling, and precipitating regimes obtained from the MIROC6 simulation and A-Train satellite observations. The spatial resolution of the reference A-Train data ($1.5° \times 1.5°$) is close to the resolution of MIROC6-SPRINTARS, which is typical of GCMs participating in model intercomparisons (e.g., CMIP6).

We obtained 74.6 million SLWCs from the model and 7.8 million SLWCs from observations. The model generated more SLWCs than were present in the A-Train observations and this suggests that one full-year of simulation with 3-hourly diagnosis is long enough to obtain robust global statistics. In the A-Train satellite retrievals, many SLWCs are located over the typical stratocumulus (Sc) regions off the west coasts of California, Peru, Australia, Namibia, and Canary (not shown), where the non-precipitating regime is dominant (Fig. 2d). The MIROC6 overestimates drizzling regimes globally by approximately 15%

(Fig. 2b). Precipitating regimes simulated by MIROC6 are in good agreement with A-Train statistics both in terms of the geographical pattern and overall occurrence (Figs. 2c and 2f).

These biases in MIROC6 can be interpreted in the context of the rain formation processes parameterized in the model. In bulk microphysics models, the onset of rain is represented by the so-called autoconversion scheme, which is generally expressed as (e.g., Berry, 1968; Beheng, 1994; Khairoutdinov and Kogan, 2000):

$$\frac{\partial q_r}{\partial t}\bigg|_{\text{aut}} = C_{\text{aut}}\, q_c^{\alpha}\, N_c^{-\beta}, \tag{3}$$

where $q_c$ and $q_r$ are the liquid cloud water and rainwater mixing ratios, respectively; $N_c$ is the cloud droplet number concentration; and $C_{\text{aut}}$, $\alpha$, and $\beta$ are the prescribed (uncertain) constants. This formulation describes how the model forms rain in terms of uncertain parameters. Given that the CloudSat cloud profiling radar is sensitive to both cloud droplets and raindrops (Stephens and Haynes, 2007; Haynes et al., 2009), model–satellite comparisons (Fig. 2) offer useful evaluations of cloud-to-

rain transition processes represented by Eq. (3), as also proposed by Kay et al. (2018).



## 3.2 Vertical microphysical structure

Figure 3 shows the CFODDs obtained from MIROC6/COSP2 and A-Train observations, which are classified according to the MODIS-derived cloud-top effective radius ($R_e$) in the 2.1 μm band as 5–12 μm, 12–18 μm, and 18–35 μm (Michibata et al., 2014). The radar reflectivity ranges ($-30$ to 20 dBZ$_e$) and the ICOD range (0 to 60) are divided linearly into 25 and 30 bins,
respectively, following Suzuki et al. (2013).

Here, we demonstrate that CFODDs deduced from satellite observations illustrate systematic transitions from non-precipitating through drizzling to precipitating regimes as a function of $R_e$, whereas MIROC6 simulates higher radar reflectivity even in the smallest $R_e$ category, revealing a "too early too frequent rain formation" bias (Suzuki et al., 2015). We attribute this discrepancy between the model and observations primarily to the following two factors: one is the bias in the updraft velocity (Nakajima
et al., 2010; Takahashi et al., 2017a) at the subgrid-scale, and the other is the uncertainty associated with the dependence of rain formation on aerosols (Wood, 2005; Suzuki et al., 2013) as characterized by $\beta$ in Eq. (3). To evaluate this regime-dependence of aerosol–cloud interactions (Sorooshian et al., 2009; Michibata et al., 2016; Chen et al., 2016, 2018), it is useful to investigate the differences in CFODDs from various environmental regimes (e.g., updraft and aerosol loading).

Thus, we defined 13 regions (Fig. 4) to examine the detailed aerosol–cloud interactions. This regional classification is based
on previous warm-rain studies with various research aims (e.g., Leon et al., 2008; Kubar et al., 2009; Sorooshian et al., 2013; Terai et al., 2015), and is summarized in Table 1. Statistics can also be examined separately over land and ocean (not shown) to investigate the differences in the CFODD transition in dynamic regimes (e.g., Takahashi et al., 2017b). Alternatively, users can define specific regions to suit their research purposes.

Figure 4 shows results from a regional CFODD analysis over five regions: Eastern Asia, Tropical Warm Pool, Equatorial
Cold Tongue, North Atlantic, and Australian. CFODDs for the smallest $R_e$ range ($5 < R_e < 12$ μm) are shown. This regional analysis reveals that the model does not always show a "too early too frequent warm-rain" bias in all regions. For example, the CFODDs over the Eastern Asia, Australian, and Equatorial Cold Tongue regions simulated by MIROC6 are in good agreement with those derived from the A-Train observations. The model accurately captures the non-precipitating regime in the smaller $R_e$ categories, suggesting that the model partially captures slower cloud-to-rain conversions in abundant-aerosol environments
(Eastern Asia) and under calm stable conditions (Australian and Equatorial Cold Tongue). These results emphasize the importance of understanding the link between microphysics and dynamics (Chen et al., 2014; Zhang et al., 2016; Michibata et al., 2016) if we wish to develop a more reliable representation of aerosol–cloud–precipitation interactions, but is beyond the scope of this technical paper.

As discussed above, CFODDs provide valuable information on cloud-to-rain microphysical transitions associated with
30 aerosol–cloud interactions and microphysics–dynamics interactions. Our new warm-rain diagnostic tool will assist in process-oriented model evaluations with the synergistic use of A-Train multi-satellite observations.





## 4 Summary

This technical paper describes a new warm-rain diagnostic tool implemented in the COSP2 satellite simulator package that extends its process-oriented diagnostic capabilities. We have introduced two new diagnostics: 1) the occurrence frequencies of non-precipitating clouds ($Z_{\max} < -15$), drizzling clouds ($-15 < Z_{\max} < 0$), and precipitating clouds ($0 < Z_{\max}$), and 2)

the PDF distributions of radar reflectivity profiles normalized by ICOD, the so-called contoured frequency by optical depth diagram (CFODD). These diagnostics make synergistic use of the CloudSat and MODIS simulators.

The diagnostic tool is controlled by the logical flags, "Lwr_occfreq" and "Lcfodd", in the namelist for COSP outputs. Users are now not required to output subcolumn parameters, such as the radar or lidar signals from simulators of active sensors, which significantly increases efficiency of model evaluation. Adding the inline warm-rain diagnostics into COSP2 increases

the computational cost only slightly (by around 0.8%) when using the SX-ACE supercomputer system of the National Institute for Environmental Studies, Japan.

The inline warm-rain diagnostic tool is intended to facilitate model evaluations that are efficient enough to be conducted within the model development loop, specifically by providing both "performance constraints" and "process-level fingerprints" (Fig. 1). The diagnostic tool has been designed to reveal potential uncertainties in modeled warm-rain processes in GCMs more

effectively by more simple way. The multi-platform products can also be extended to include other diagnostics for mixed-phase and ice clouds in future work. Requests for specific diagnostics, particularly those requiring COSP subcolumn output for fast process evaluations, are welcomed.



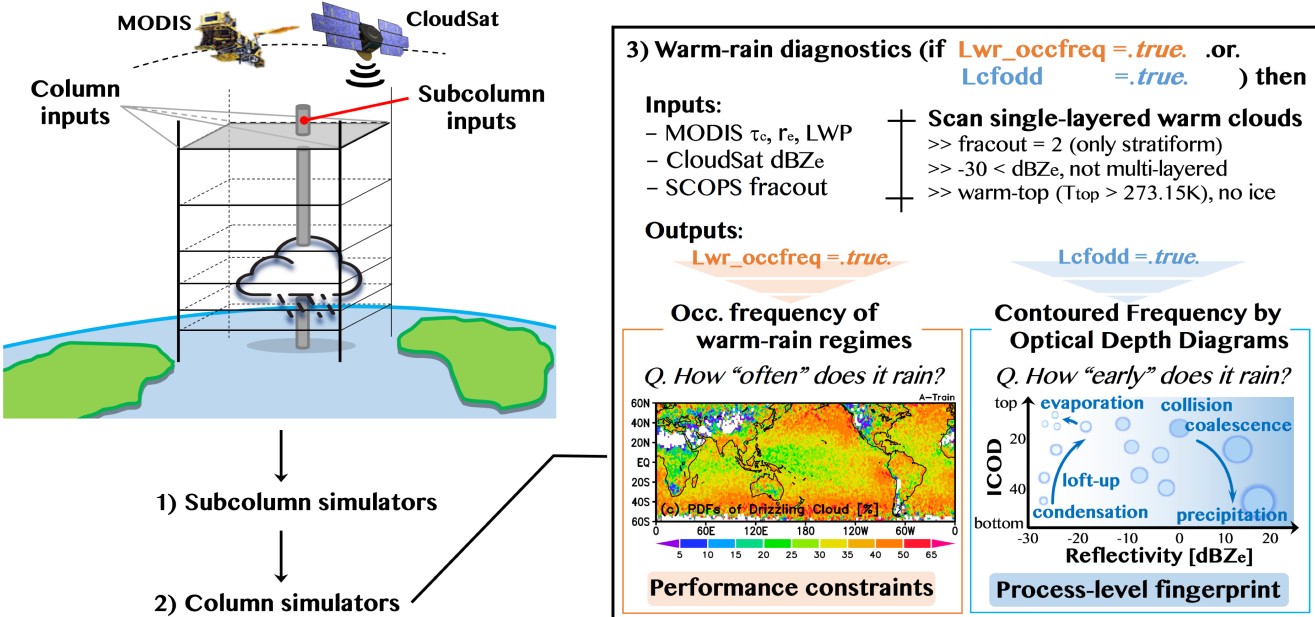

**Figure 1.** Schematic flowchart of COSP2 (see also Swales et al. (2018) for details) and additional processes for warm-rain diagnostics introduced in this work.



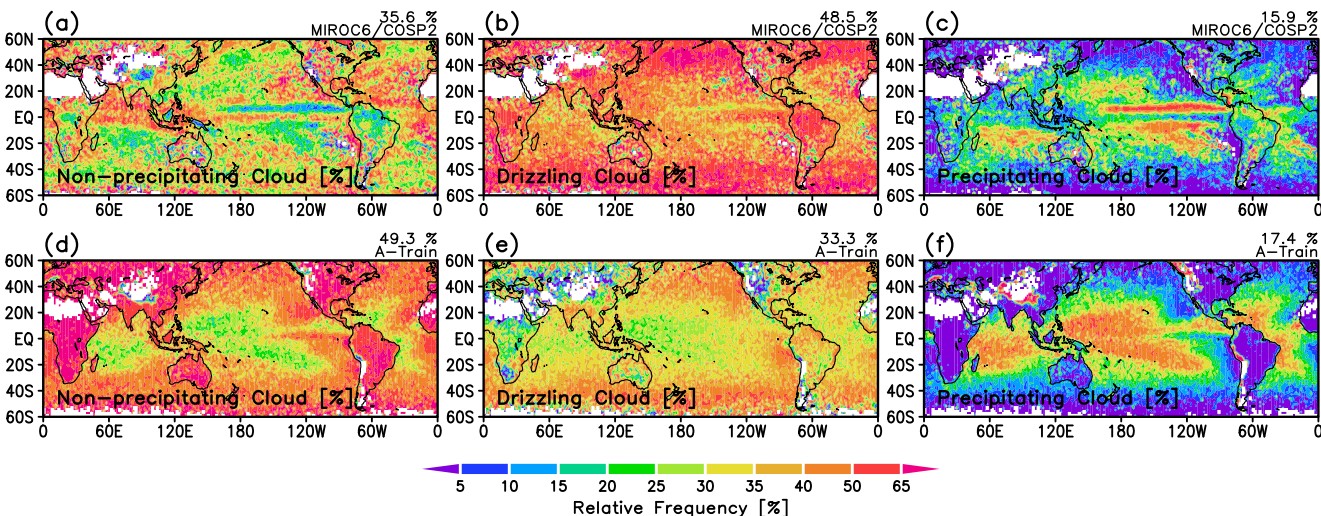

**Figure 2.** Geographical maps of the fractional occurrence of (a, d) non-precipitating clouds ($Z_{\max} < -15$), (b, e) drizzling clouds ($-15 < Z_{\max} < 0$), and (c, f) precipitating clouds ($0 < Z_{\max}$) obtained from (top) the MIROC6/COSP2 one full-year simulation, and (bottom) the A-Train satellite observations for the period June 2006–April 2011. Global means of the occurrence frequency are shown at the top right of each panel.





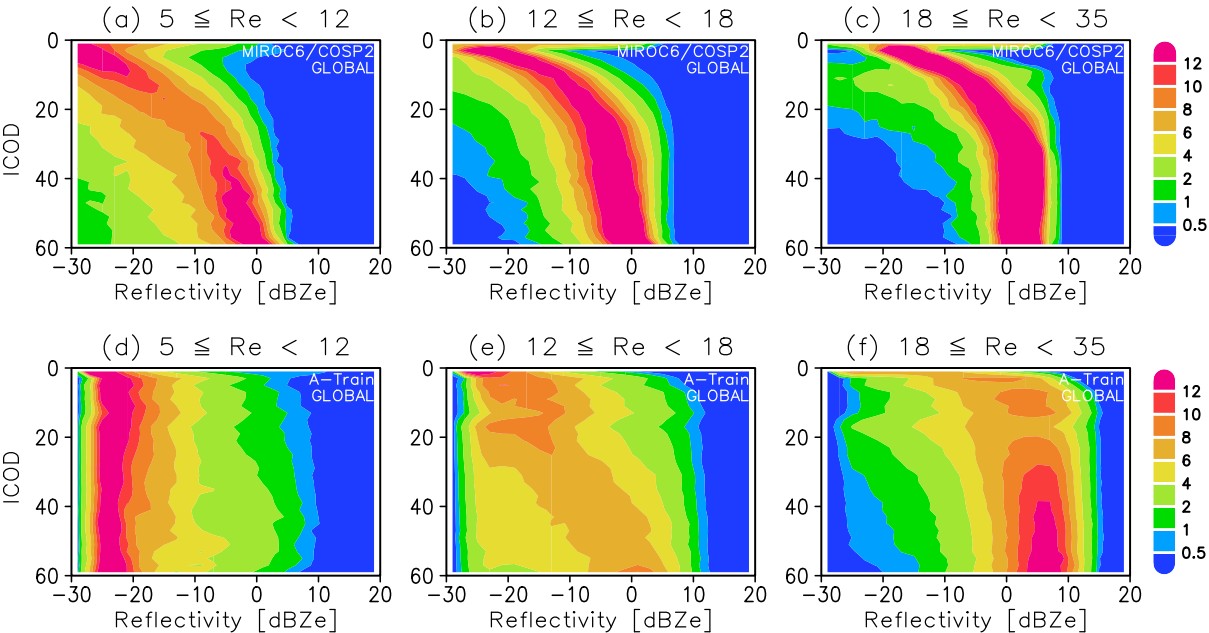

**Figure 3.** Contoured frequency by optical depth diagrams (CFODDs) obtained from (top) the MIROC6/COSP2 one full-year simulation, and (bottom) the A-Train satellite observations for the period June 2006–April 2011. CFODDs are classified according to the MODIS-derived cloud-top effective radius ($R_e$) in the 2.1 μm band as (a, d) 5–12, (b, e) 12–18, and (c, f) 18–35 μm following Michibata et al. (2014).



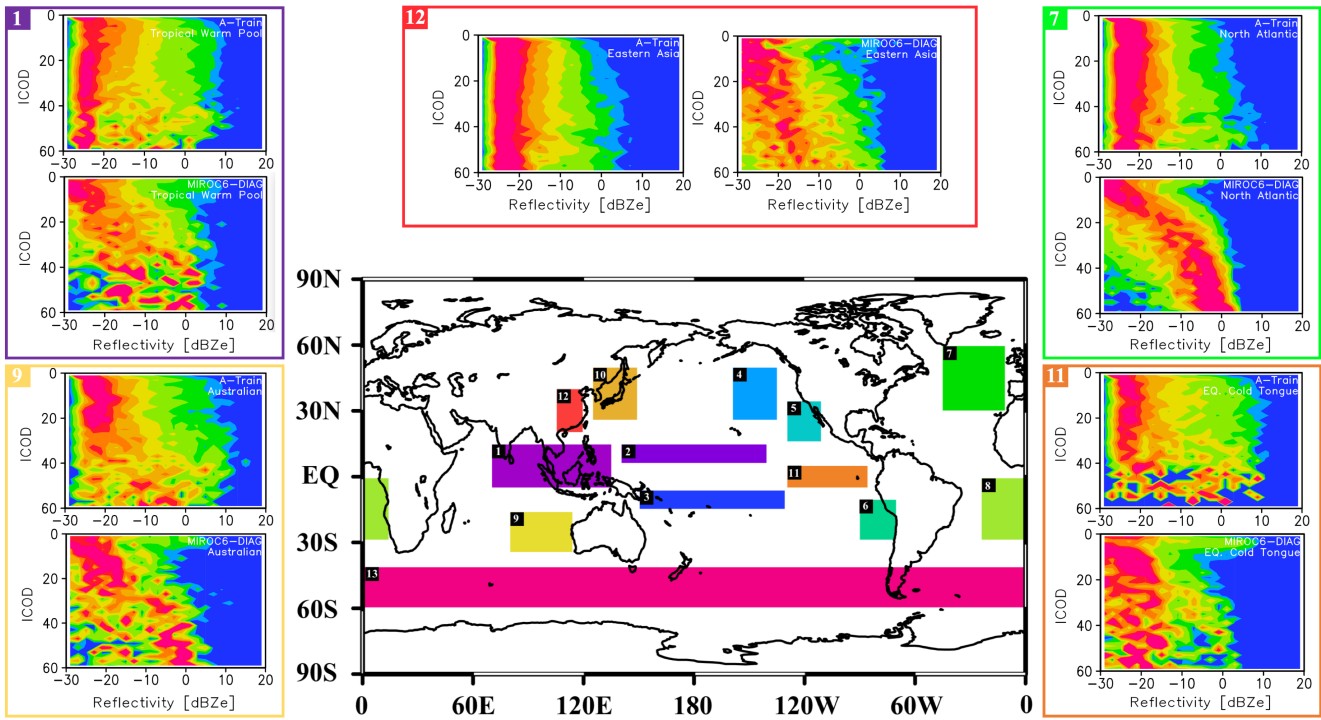

**Figure 4.** Definition of the 13 regions used in the post-process package. An example of the regional CFODDs analysis over the (red) Eastern Asia, (purple) Tropical Warm Pool, (yellow) Australian, (green) North Atlantic, and (orange) Equatorial Cold Tongue regions, obtained from the MIROC6/COSP2 and the A-Train observations for the $R_e$ range $5 < R_e < 12 \, \mu$m.



**Table 1.** Definition of the 13 regions used in the CFODD regional analysis, corresponding to the boxes in Figure 4.

|  | Region | Latitude, Longitude |
|---|---|---|
| 1) | Tropical Warm Pool | 5°S–20°N, 70°E–150°E |
| 2) | ITCZ | 5°N–15°N, 140°E–140°W |
| 3) | SPCZ | 15°S–5°S, 150°E–130°W |
| 4) | North East Pacific | 25°N–50°N, 160°W–135°W |
| 5) | California StCu deck | 15°N–35°N, 140°W–110°W |
| 6) | Peruvian | 30°S–0°S, 120°W–70°W |
| 7) | North Atlantic | 30°N–60°N, 45°W–10°W |
| 8) | Namibian | 30°S–0°S, 25°W–15°E |
| 9) | Australian | 40°S–15°S, 60°E–115°E |
| 10) | Japan | 25°N–50°N, 125°E–150°E |
| 11) | Eqt. Cold Tongue | 5°S–5°N, 130°W–85°W |
| 12) | Eastern Asia | 20°N–40°N, 100°E–120°E |
| 13) | Southern Ocean | 40°S–60°S |

*Code and data availability.*  The source code for COSP2 is available from a GitHub repository (https://github.com/CFMIP/COSPv2.0). Post-processing code and reference A-Train statistics are available from a Zenodo repository (to be opened after the acceptance of this manuscript). The results of the MIROC6 simulation used to produce the figures are also included in the post-process package. The source code and data can be provided to the reviewers for the purpose of reviewing the manuscript.

*Author contributions.*  T.M. and K.S. designed the research; T.M. implemented the new diagnostics into COSP2; T.M., T.O., and X.J. tested the new tool on the MIROC6/COSP2 interface; K.S. improved and reviewed the code; T.M. analyzed the model and observational data; and T.M. and K.S. wrote the paper.

*Competing interests.*  The authors declare that they have no conflict of interest.

*Acknowledgements.*  We thank the developers of the COSP satellite simulator and the CFMIP community. This work has benefited from
fruitful discussions with Alejandro Bodas-Salcedo, Robert Pincus, and Dustin Swales. Simulations by MIROC-SPRINTARS were executed on the SX-ACE supercomputer system of the National Institute for Environmental Studies, Japan. The MODIS Collection 6 products are available from the LAADS website (https://ladsweb.nascom.nasa.gov). The CloudSat data products were provided by the CloudSat Data Processing Center at CIRA/Colorado State University (http://www.cloudsat.cira.colostate.edu). This study was supported by the JSPS KAK-ENHI Grant Numbers JP18J00301 and JP19K14795; the Integrated Research Program for Advancing Climate Models (TOUGOU program)
from the Ministry of Education, Culture, Sports, Science and Technology (MEXT), Japan; and the Collaborative Research Program of the





Research Institute for Applied Mechanics, Kyushu University. K.S. was supported by NOAA's Climate Program Office's Modeling, Analysis, Predictions, and Projections program with grant number NA15OAR4310153.





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
