# Peer review of "Incorporation of inline warm rain diagnostics into the COSP2 satellite simulator for process-oriented model evaluation"

_Geoscientific Model Development, 2019_

## Referee Comment (RC1) · Anonymous Referee #1 · 4 Jul 2019

The authors describe enhancements to the COSP satellite simulator package intended to aid in model development and evaluation of precipitation processes. New diagnostics on subcolumn fields permit better and easier comparison with satellite datasets, and the paper provides an excellent example of this capability using A-Train data.

This paper is clearly written, concise, well referenced, and should provide a useful guide for users of these tools in the future. I found it well organised and clear, and have only minor recommendations for ways to improve the manuscript, which are mostly textual in nature. It suits the scope of this journal and should be suitable for publication after some minor modifications.

[Figure]

Minor points:

Title and throughout – The authors use 'warm-rain' consistently with a hyphen, whereas elsewhere in the literature it is usually not hyphenated (e.g. Stephens and Haynes 2007, Chen et al. 2011, Suzuki et al. 2011). I would suggest changing this to match the literature. In the specific case of the title, 'warm-rain' could be replaced by 'precipitation.'

P1L1 - 'has been widely used' contains a value judgment, and could be simplified to 'is used'

P3L2 – This is one example of this manuscript's slight tendency for overdoing the number of citations in some places. Here and elsewhere it may be better for readers if the authors select one or two of the most important and relevant citations, rather than a half dozen.

P3L10 – I found the beginning of this section to be quite abrupt, immediately discussing CFADs without putting this into context. Given that the section is titled 'Concept and design' it might be useful for readers if the overall concept is stated before going straight into the details. Perhaps a short paragraph before Section 2.1 begins, or a sentence to lead into why CFADs are then discussed. This is done in a roundabout way later in this first paragraph, implying that such diagnostics are useful for fingerprinting processes. It would read better if this were rearranged a little.

P4L12 – 'A-Train' should be defined, either here or at its first mention (P3L6), preferably with a citation to one of the relevant Stephens or L'Ecuyer papers.

P4L28 – This section title might be better without 'scientific perspectives' in there, as this is quite vague.

P4L31 – Stay consistent, either refer to COSP or COSP2.

P5L6 – Please state which specific data products were used including which version or processing level, as this is more important for readers' interpretation than the original

papers cited here.

P5L16 – I disagree with the causative statement here, saying that because the model generated more SLWCs this means that the chosen period is good enough for robust statistics. This could be rewritten to say that it is indeed a long enough period (which I agree with), but that is not proven by the fact that the model had more SLWCs, which is what the current text suggests.

P5L19 – This could be clearer, as saying that 'MIROC6 overestimates . . . by 15%' can be misleading. Please say what the reference is, or it might be best if just stating that MIROC6 finds 48.5% drizzle versus 33.3% in A-Train data. What I mean is, if the A-Train data are taken as truth, then MIROC6 overestimates drizzle by almost 50% relative to the A-Train data, so it's better to write what is meant explicitly so that it can't be misinterpreted.

P5L21 – I disagree with the authors' interpretation that the model's 'geographical pattern' of precipitation is in good agreement with observations. I would suggest that this statement should be modified or given some caveats at least, since the patterns in the tropical oceans and continental precipitation seem quite different in the figure.

P6L6 – It is implied here that effective radius and the subdivisions of Re used in the analysis are related to whether or not clouds are precipitating. This is surely spelled out in some of the referenced literature, but a sentence or two stating this explicitly would be useful for readers. A reference to Lebsock et al. 2008 might be helpful.

P7L15 – Reword 'by more simple way'

Fig 1 – The use of double quotes to show emphasis (How "often" does it rain) should be replaced by a switch to regular/italic font, or removed.

Fig 4 – Please state in the caption whether the colour scale used is identical to those in Fig 3. If not, please provide a colour bar in the figure.

[Figure]

---

## Referee Comment (RC2) · Anonymous Referee #2 · 15 Jul 2019

Review Michibata

Michibata et al. propose new inline warm rain diagnostics for a community satellite simulator project (COSP) to enable process-oriented model evaluation. The diagnostics are well described and motivated. The diagnostics are similar to those developed by Kay et al. (2018) for precipitation frequency, but include important process-inspired criteria beyond Kay et al. (2018) for warm rain. The implementation of these new diagnostics in MIROC6 reveals interesting results. I have only minor comments. I recommend publication.

Abstract. Can you more specifically describe the diagnostics beyond "two diagnostics

for warm rain processes"? What are the new outputs in COSP?

Line 5-7, page 2. I recommend removing "apple-to-orange" and "apple-to-apple". Instead - can you describe what makes the comparisons more credible when they are done with COSP?. For example, the reader should be aware that "definition aware" and "scale aware" comparisons are made possible by the use of satellite simulators and a sub-column generator respectively.

Line 12, page 2. "such as CMIP6" to "including CMIP6" or "e.g., CMIP6". Much was done with CMIP5 as well...

Line 30-31, page 4. Please explain how the number of sub-columns (140) was selected. The rationale behind the selection of the number of sub-columns is important to describe, especially for those new to COSP.

Line 12-14, page 5. "The spatial resolution of the reference A-train data ...". This sentence is incorrect. The A-train native spatial resolution is much higher than 1.5 degrees – For CloudSat it is ∼1 km. While the statistics of both the models and the observations are compared at 1.5 degrees – this study has taken a lot more care to make "scale aware" comparison not at 1.5 degrees. Specifically, the climate model data were "down-scaled" to the A-train data native resolution using a sub-column generator in COSP. Please describe in detail so that the reader does not confuse the resolution of the grid at which the statistics are reported (1.5 degrees) and the resolution at which the comparisons are being made («« < 1.5 degrees).
* * *

---

## Referee Comment (RC3) · Johannes Mülmenstädt (Referee) · 31 Jul 2019

I have reviewed "Incorporation of inline warm-rain diagnostics into the COSP2 satellite simulator for process-oriented model evaluation" by Michibata et al. The manuscript documents extensions to the COSP v2 satellite simulator package that, in my opinion, will greatly advance the understanding of warm rain processes in GCMs and contribute to improvements in process realism.

Below is a list of fairly minor comments that should be addressed before publication.

- I am not complaining about the $-15$ and 0 dBZe thresholds (they seem to be

used frequently), but I would appreciate a sentence of discussion or a reference on why these particular values were chosen.

- Similarly, I am not complaining about the use of simple, column-maximum $Z_e$, but perhaps the authors could comment on the advantages and disadvantages of this approach compared to the CloudSat precipitation flag simulator presented in Kay et al. (2018).

- I believe the recommendation of 100 subcolumns per one degree (lat/lon) of model resolution deserves explanation or a reference.

- In the same vein, it appears that the authors intend for the number of subcolumns to scale with the grid spacing, not the grid-box area (140 subcolumns at 1.4 degree resolution, p. 4, l. 31). It would be good to explain why.

- In the discussion of the model results, the authors should explain whether their CFODDs include convective precipitation or only stratiform, and whether MIROC uses the same microphysics in convective and stratiform clouds.

- Please confirm that one of the repositories listed in the code availability section will contain the source code for the online statistics (I assume so, but the wording is a bit ambiguous).

I am also attaching an annotated PDF with very minor comments that the authors may find helpful in proofreading the manuscript.

Please also note the supplement to this comment:
https://www.geosci-model-dev-discuss.net/gmd-2019-104/gmd-2019-104-RC3-supplement.pdf

**Supplement:**

[revised manuscript text omitted]

---

## Short Comment (SC1) · 14 Aug 2019

This comment is written to raise respects in which this manuscript is not compliant with GMD policy. The issues raised here need to be addressed before any revised manuscript can be accepted.

[Figure]

**Code and data embargo**

It is not acceptable to embargo code or data from a GMDD manuscript. This undermines the open peer review process. The Zenodo archive of the data and scripts needs to be immediately published (for example by citing it in a response to this comment) in order to enable readers of the GMDD manuscript to properly review the work.

**Code on GitHub**

The reference to the COSP2 code is a GitHub link. While GitHub is an excellent development platform, it is not a suitable archive location. Indeed, GitHub themselves tell you to use Zenodo for this purpose and provide integration to make this easy[1]. Please produce a suitable archive of the code (e.g. on Zenodo) and cite this.

For further details, including the absolute prohibition on embargoes, please see the GMD model code and data policy[2]
* * ** * *
[1]https://guides.github.com/activities/citable-code/

[2]https://www.geoscientific-model-development.net/about/code_and_data_policy.html

---

## Author Comment (AC1) · 2 Sep 2019

**Response to Reviewer #1 of gmd-2019-104**

Dear Reviewer #1,

Thank you very much for taking your time to review our paper. We think that your comments greatly help improve the manuscript. We have revised the manuscript according to your comments as explained below with point-by-point responses to your comments. We hope that the revision is enough to address your comments to make the manuscript now acceptable for publication in *GMD*.

**[RC]:** *Referee comment*
**[AC]:** **Author comment**

**Reviewer #1:**

**[RC]** *The authors describe enhancements to the COSP satellite simulator package intended to aid in model development and evaluation of precipitation processes. New diagnostics on subcolumn fields permit better and easier comparison with satellite datasets, and the paper provides an excellent example of this capability using A-Train data.*
*This paper is clearly written, concise, well referenced, and should provide a useful guide for users of these tools in the future. I found it well organised and clear, and have only minor recommendations for ways to improve the manuscript, which are mostly textual in nature. It suits the scope of this journal and should be suitable for publication after some minor modifications.*
**[AC]** We would like to thank the referee #1 for his/her positive and constructive comments. The reply and corrections on individual comments are below. Note here that, page and line numbers denoted in the authors' responses below correspond to the track-changes file, not original manuscript.

**Minor points:**

**[RC1]** *Title and throughout – The authors use 'warm-rain' consistently with a hyphen, whereas elsewhere in the literature it is usually not hyphenated (e.g. Stephens and Haynes 2007, Chen et al. 2011, Suzuki et al. 2011). I would suggest changing this to match the literature. In the specific case of the title, 'warm-rain' could be replaced by 'precipitation.'*
**[AC1]** We have removed the hyphen, and unified as "warm rain" throughout the manuscript. The title still uses "warm rain" not "precipitation", because the present study is limited to liquid-phase cloud microphysics and inline diagnostics for mixed- and ice-phase cloud microphysics are also planned to be introduced as future work, as described in section 4.

**[RC2]** *P1L1 - 'has been widely used' contains a value judgment, and could be simplified to 'is used'*
**[AC2]** We have modified, thanks.

**[RC3]** *P3L2 – This is one example of this manuscript's slight tendency for overdoing the number of citations in some places. Here and elsewhere it may be better for readers if the authors select one or two of the most important and relevant citations, rather than a half dozen.*
**[AC3]** We agree with the reviewer to avoid citing too many papers. Accordingly, we have selected the most important citations, throughout the revised manuscript.
Page 2, Line 2: removed Wood (2012).
Page 2, Line 8: removed Michibata et al. (2016); Bai et al. (2018).
Page 2, Line 25: added Maloney et al. (2019)
Page 3, Line 15: removed Tompkins and Di Giuseppe (2015), Norris and da Silva (2016), Griffin and Larson (2016), and Ovchinnikov et al. (2016).

Page 5, Line 1: removed Bennartz (2007).
Page 5, Line 23: added Michibata et al. (2019).
Page 7, Line 28: removed Kubar et al. (2009); Sorooshian et al. (2013).
Page 8, Line 7: removed Michibata et al. (2016).
Page 8, Line 27: added Mülmenstädt et al. (2015); Kikuchi et al. (2017)

**[RC4]** *P3L10 – I found the beginning of this section to be quite abrupt, immediately discussing CFADs without putting this into context. Given that the section is titled 'Concept and design' it might be useful for readers if the overall concept is stated before going straight into the details. Perhaps a short paragraph before Section 2.1 begins, or a sentence to lead into why CFADs are then discussed. This is done in a roundabout way later in this first paragraph, implying that such diagnostics are useful for fingerprinting processes. It would read better if this were rearranged a little.*
**[AC4]** We agree that this paragraph requires clarification, and have rearranged sections 2 and 2.1 in the revised manuscript.
Section 2: added overall concept in this work as follows:
"The objective of this work is to provide a specific "process-oriented" metrics that is also compatible with "scale-aware" and "definition-aware" diagnostics (Kay et al., 2018) in the manner implemented into COSP for fair comparison of warm clouds among GCMs and satellite retrievals. Here the main concept is using conditional statistics that "fingerprint" the process of interest, by combining multiple satellite observables. One of the transformative advances recently made possible by combining active and passive satellite measurements is the ability to generate observational diagnostics of how the microphysical vertical structure of clouds varies with the surrounding environment (Marchand et al., 2009; Sorooshian et al., 2013), such as aerosol concentration (Ma et al., 2018; Rosenfeld et al., 2019) and dynamical regimes (Nam et al., 2014; Christensen et al., 2016).

As a default diagnostic from the CloudSat radar simulator alone in COSP (Bodas-Salcedo et al., 2011), the so-called contoured frequency by altitude diagram (CFAD) is prepared to provide macrophysical vertical structure including all types of hydrometeors (i.e., liquid droplets, ice crystals, raindrops, and snowflakes). In this regard, more specific statistics are useful when investigating a particular process, including the warm rain microphysical processes that are the focus of this work as described below."
Section 2.1: In accordance with the response above, some sentences were rearranged.

**[RC5]** *P4L12 – 'A-Train' should be defined, either here or at its first mention (P3L6), preferably with a citation to one of the relevant Stephens or L'Ecuyer papers.*
**[AC5]** We have changed the sentence at Page 2 Line 9 from 'Global satellite observations, particularly those of satellite constellations' to 'The A-Train global observations (Stephens et al., 2002; L'Ecuyer and Jiang, 2010), consisting of the sun-synchronous and polar-orbiting multisatellite constellation, …'.

**[RC6]** *P4L28 – This section title might be better without 'scientific perspectives' in there, as this is quite vague.*
**[AC6]** We have changed the section title to "Examples of model–observation intercomparisons".

**[RC7]** *P4L31 – Stay consistent, either refer to COSP or COSP2.*
**[AC7]** The revised manuscript uses "COSP", except the case that the sentence does not mean the latest version of COSP.
Page 5 Line 29: "COSP2" to "COSP"
Page 8 Line 20: "COSP2" to "COSP"

**[RC8]** *P5L6 – Please state which specific data products were used including which version or processing level, as this is more important for readers' interpretation than the original papers cited here.*
**[AC8]** We have added more detailed description for the A-Train datasets used as follows.
Page 6 Line 3: "As a reference, we also calculated the target metrics (i.e., the occurrence frequency of SLWCs and CFODDs) using CloudSat and MODIS satellite data products (e.g., Stephens et al., 2008)

for the period June 2006–April 2011. The visible cloud optical depth and 2.1 µm cloud droplet effective radius were derived from MODIS level 2B-TAU R04 product (Polonsky, 2008), radar reflectivity profile was obtained from CloudSat-derived level 2B-GEOPROF R04 product (Mace et al., 2007; Marchand et al., 2008), and the pressure and temperature profiles were derived from the ECMWF-AUX R04 product (Partain, 2007). Detailed descriptions of the model configuration and the analysis procedure to detect SLWCs are provided elsewhere (Michibata and Takemura, 2015; Michibata et al., 2016).".

**[RC9]** *P5L16 – I disagree with the causative statement here, saying that because the model generated more SLWCs this means that the chosen period is good enough for robust statistics. This could be rewritten to say that it is indeed a long enough period (which I agree with), but that is not proven by the fact that the model had more SLWCs, which is what the current text suggests.*
**[AC9]** Yes, we agree to the comment. This sentence has been modified as follows: "… present in the A-Train observations. This suggests that one full-year of simulation with 3-hourly diagnosis is long enough, but note that this does not negate the possibility of too frequent generation of SLWCs in the model.".

**[RC10]** *P5L19 – This could be clearer, as saying that 'MIROC6 overestimates … by 15%' can be misleading. Please say what the reference is, or it might be best if just stating that MIROC6 finds 48.5% drizzle versus 33.3% in A-Train data. What I mean is, if the A-Train data are taken as truth, then MIROC6 overestimates drizzle by almost 50% relative to the A-Train data, so it's better to write what is meant explicitly so that it can't be misinterpreted.*
**[AC10]** Thank you for suggestion. This sentence has been modified as follows: "The MIROC6 finds 48.5% drizzling regime versus 33.3% in the A-Train retrievals (Figs. 2b and 2e).".

**[RC11]** *P5L21 – I disagree with the authors' interpretation that the model's 'geographical pattern' of precipitation is in good agreement with observations. I would suggest that this statement should be modified or given some caveats at least, since the patterns in the tropical oceans and continental precipitation seem quite different in the figure.*
**[AC11]** We agree with the reviewer. The geographical pattern simulated in MIROC6 is different from observations mainly over the tropical oceans and continents. We have noted the model bias in the revised manuscript as follows: "For precipitating regime, although the global mean values of occurrence frequency are consistent with each other (15.9% in MIROC6 and 17.4% in A-Train), the geographical pattern is quite different particularly over tropical oceans and continents (Figs. 2c and 2f), implying that the model has biases in the warm rain formation process (e.g., Jing et al., 2019) and/or the representation of cloud types (e.g., Huang et al., 2015).".
Page 7 Line 3: changed "rain formation processes" to "aerosol–cloud interactions".

**[RC12]** *P6L6 – It is implied here that effective radius and the subdivisions of Re used in the analysis are related to whether or not clouds are precipitating. This is surely spelled out in some of the referenced literature, but a sentence or two stating this explicitly would be useful for readers. A reference to Lebsock et al. 2008 might be helpful.*
**[AC12]** This sentence has been changed as follows: "… regimes as a function of Re, and is consistent with previous observational findings that showed the strong dependence of the onset of precipitation upon Re (Lebsock et al., 2008; Rosenfeld et al., 2012). On the other hand, MIROC6 simulates higher …".

**[RC13]** *P7L15 – Reword 'by more simple way'*
**[AC13]** This has been changed to "… in GCMs more effectively and simply.".

**[RC14]** *Fig 1 – The use of double quotes to show emphasis (How "often" does it rain) should be replaced by a switch to regular/italic font, or removed.*

**[AC14]** The double quotes in Fig. 1 have been removed in the revised version. We have also corrected the typo in sampling method in this figure (and also Page 5 Line 9), "fracout = 2" to "fracout = 1", which means stratiform cloud type. The results analyzed are not changed.

**[RC15]** *Fig 4 – Please state in the caption whether the colour scale used is identical to those in Fig 3. If not, please provide a colour bar in the figure.*
**[AC15]** The color scale is the same as in Fig. 3. This has been noted in the caption in Fig. 4.

Thank you very much again for reviewing our paper.

Sincerely yours,

Takuro Michibata

---

## Author Comment (AC2) · 2 Sep 2019

**Response to Reviewer #2 of gmd-2019-104**

Dear Reviewer #2,

Thank you very much for taking your time to review our paper. We think that your comments greatly help improve the manuscript. We have revised the manuscript according to your comments as explained below with point-by-point responses to your comments. We hope that the revision is enough to address your comments to make the manuscript now acceptable for publication in *GMD*.

**[RC]**: *Referee comment*
**[AC]**: **Author comment**

**Reviewer #2:**

**[RC]** *Michibata et al. propose new inline warm rain diagnostics for a community satellite simulator project (COSP) to enable process-oriented model evaluation. The diagnostics are well described and motivated. The diagnostics are similar to those developed by Kay et al. (2018) for precipitation frequency, but include important process-inspired criteria beyond Kay et al. (2018) for warm rain. The implementation of these new diagnostics in MIROC6 reveals interesting results. I have only minor comments. I recommend publication.*
**[AC]** We would like to thank the referee #2 for his/her positive comments. We have revised the manuscript according to the referee comments as listed below. Note here that, page and line numbers denoted in the authors' responses below correspond to the track-changes file, not original manuscript.

**Specific points:**

**[RC1]** *Abstract. Can you more specifically describe the diagnostics beyond "two diagnostics for warm rain processes"? What are the new outputs in COSP?*
**[AC1]** We have added more specific description about the two diagnostics as follows: "Here, we incorporate two diagnostics for warm rain microphysical processes into the latest version of the simulator (COSP2). The first one is the occurrence frequency of warm rain regimes (i.e., non-precipitating, drizzling, and precipitating) classified according to CloudSat radar reflectivity, putting the warm rain process diagnostics into the context of geographical distributions of precipitation. The second diagnostic is the probability density function of radar reflectivity profiles normalized by the in-cloud optical depth, the so-called contoured frequency by optical depth diagram (CFODD), which illustrates how the warm rain processes occur in vertical dimension using statistics constructed from CloudSat and MODIS simulators."
In accordance with the rearranged above, the sentence was modified as follows: "… produce statistics online along with the subcolumn information during the COSP execution, …".

**[RC2]** *Line 5-7, page 2. I recommend removing "apple-to-orange" and "apple-to-apple". Instead - can you describe what makes the comparisons more credible when they are done with COSP? For example, the reader should be aware that "definition aware" and "scale aware" comparisons are made possible by the use of satellite simulators and a sub-column generator respectively.*
**[AC2]** Thank you for suggestion. We removed "apple-to-orange" and "apple-to-apple". Instead, we have modified the sentence that explains the merit of COSP as follows:
Page 2 Line 15: "… same algorithms applied to each satellite sensor for consistent ("definition-aware") comparison. Furthermore, the process evaluation among models and observations should be done under the same spatiotemporal scale for consistent ("scale-aware") comparison."

Page 2 Line 33: "… from multiple instrument simulators in a "definition-aware and scale-aware" framework (Kay et al., 2018)."

**[RC3]** *Line 12, page 2. "such as CMIP6" to "including CMIP6" or "e.g., CMIP6". Much was done with CMIP5 as well…*
**[AC3]** We have modified the sentence in the revised manuscript.

**[RC4]** *Line 30-31, page 4. Please explain how the number of sub-columns (140) was selected. The rationale behind the selection of the number of sub-columns is important to describe, especially for those new to COSP.*
**[AC4]** For CFMIP2 experiments, COSP users are recommended to assume ~100 subcolumns per 1 degree of model grid spacing to enable comparison to satellite sampling at the kilometer scale, as described in Page 3 Line 7. This statement is based on the README file in CFMIP2 experiments for recommended configuration (cfmip2/cosp_input_cfmip2_long_inline.txt). The model horizontal resolution is T85 (~1.4 degree in longitude and latitude) in this study, and hence we prepared 140 subcolumns in COSP execution. We have added this note in the revised manuscript.

**[RC5]** *Line 12-14, page 5. "The spatial resolution of the reference A-train data …". This sentence is incorrect. The A-train native spatial resolution is much higher than 1.5 degrees – For CloudSat it is ~1 km. While the statistics of both the models and the observations are compared at 1.5 degrees – this study has taken a lot more care to make "scale aware" comparison not at 1.5 degrees. Specifically, the climate model data were "down-scaled" to the A-train data native resolution using a sub-column generator in COSP. Please describe in detail so that the reader does not confuse the resolution of the grid at which the statistics are reported (1.5 degrees) and the resolution at which the comparisons are being made («« < 1.5 degrees).*
**[AC5]** Thank you for such an important comment, and we agree with the reviewer. The following notes have been added in the revised manuscript: "Note that although the reference A-Train statistics is shown at 1.5º x 1.5º resolution, which is close to that of MIROC6-SPRINTARS, the statistics are constructed from the native CloudSat resolution (1.4 x 2.5 km) and subcolumns in the host model prepared by COSP (kilometer scale) to achieve the "scale-aware" model-satellite comparison.".

Thank you very much again for reviewing our paper.

Sincerely yours,

Takuro Michibata

---

## Author Comment (AC3) · 2 Sep 2019

The comment was uploaded in the form of a supplement:
https://www.geosci-model-dev-discuss.net/gmd-2019-104/gmd-2019-104-AC3-supplement.pdf

---

## Author Comment (AC4) · 2 Sep 2019

**Response to Short Comment of gmd-2019-104**

Dear Dr. David Ham,

Thank you very much for posting the comment on the discussion forum. I am returning herewith a manuscript revised according to the comment.

**[SC]**: *Short comment*
**[AC]**: **Author comment**

**David Ham (Executive editor):**

**[SC]** *This comment is written to raise respects in which this manuscript is not compliant with GMD policy. The issues raised here need to be addressed before any revised manuscript can be accepted.*

***Code and data embargo***
*It is not acceptable to embargo code or data from a GMDD manuscript. This undermines the open peer review process. The Zenodo archive of the data and scripts needs to be immediately published (for example by citing it in a response to this comment) in order to enable readers of the GMDD manuscript to properly review the work.*

***Code on GitHub***
*The reference to the COSP2 code is a GitHub link. While GitHub is an excellent development platform, it is not a suitable archive location. Indeed, GitHub themselves tell you to use Zenodo for this purpose and provide integration to make this easy[1]. Please produce a suitable archive of the code (e.g. on Zenodo) and cite this.*
*For further details, including the absolute prohibition on embargoes, please see the GMD model code and data policy[2]*
*[1]https://guides.github.com/activities/citable-code/*
*[2]https://www.geoscientific-model-development.net/about/code_and_data_policy.html*

**[AC]** We would like to thank Dr. David Ham for his notification of data policy in GMD. The source code of COSP2 for the online diagnostics[1] and data[2] used in this study are included in the Zenodo repository, and are now open to public. We have modified the 'Code and data availability' section in the revised manuscript.
[1] https://doi.org/10.5281/zenodo.1442468
[2] https://doi.org/10.5281/zenodo.3370823

Thank you very much for handling our paper.

Sincerely yours,

Takuro Michibata